# Carbon subsurface traffic jam as driver for methane oxidation activity and selectivity on palladium surfaces

Ulrike Küst [1,2] ✉, Rosemary Jones[1,3], Julia Prumbs [1], Alessandro Namar [4], Mattia Scardamaglia [3], Andrey Shavorskiy [3] & Jan Knudsen [1,2,3] ✉

Separating how surface and subsurface species affect catalytic function is a challenging task in heterogeneous catalysis, particularly when deposition and segregation take place at reaction conditions. Here, we report on an operando approach to establish surface/subsurface/function correlations. Using temperature modulations we oscillate carbon deposition and segregation over a Pd catalyst. Catalytic composition and function are monitored during methane oxidation showing that the surface coverage of carbon drives partial oxidation to CO, while subsurface carbon controls the overall methane turnover. Also, we show that a carbon traffic jam in the subsurface leads to a shifting selectivity from $H_2$ to $H_2O$ formation, highlighting the importance of the catalyst subsurface for the catalytic reaction.

An often-used simplification in heterogeneous catalysis research is that the bulk of the catalyst is frozen and only the surface structure changes at reaction conditions[1]. For this reason, the common approach is to correlate surface structure with catalytic function.

An increasing amount of evidence points, however, towards the importance of the subsurface region a few nanometers below the surface, which can play a decisive role in controlling catalytic function[1,2]. For example, the controlled doping of subsurface layers can be used to enhance catalytic performance[3–6]. However, not all subsurface changes are controllable. If a reaction intermediate dissolves into the subsurface layers[7] it is easy to overlook its effect on catalytic properties. The most well-known examples involve hydrogen[1,8,9], oxygen[2,10], and especially carbon[1,8,11–13] dissolution. Additionally, the deposition and subsequent bulk-dissolution of these species can even lead to continuous (structural) changes of the catalyst surface[7,14–19], and subsurface[7] under reaction conditions. However, studies that discuss this dynamic nature of both surface and subsurface, for example in the case of carbon deposition and dissolution are rare. This is unfortunate, particularly since carbon is well-known to impact all aspects of catalytic function[20–22].

One catalyst that can suffer from carbon deposition is Pd. This can happen when it is used to catalyze methane oxidation and dynamic studies of this catalyst and reaction would therefore be interesting[23–25]. The majority of recently published $CH_4$ oxidation studies focuses, however, on complete oxidation of methane during which no carbon deposition is observed and the only reaction product is $CO_2$[26–34], as for example discussed in a recent account summarizing work from groups at Standford, ETH Zürich, and the Paul Scherrer Institute (PSI)[26]. Some studies mentioned there discuss the oxidation of Pd foils and $Al_2O_3$ supported Pd particles when exposed to oxygen-rich to stochiometric $CH_4:O_2$ gas mixtures at mbar conditions and while the catalyst is active[27,35]. On these catalysts, the poisoning effect of water observed at temperatures below 450°C has been studied[26,36] and it has been demonstrated how dynamic operation consisting of short reducing pulses can be used to prevent steam-induced sintering[37]. Another example of advantageous dynamic operation is the recent study by Roger et al.[38] where the frequency and amplitude of $O_2$ pulses were used to enhance the activity of $CH_4$ oxidation over a Pd/$Al_2O_3$ catalyst. Besides using dynamic operations for achieving better catalyst function, it is also frequently used in so-called Modulation Excitation Spectroscopy (MES)[39–41] for identifying minority phases[42], for studying catalysts at off-equilibrium conditions, or for deconvoluting reaction kinetics of reactions running in parallel. An example of the latter for $CH_4$ oxidation is a study by Marchionni et al.[43], in which alternating

[1]Division of Synchrotron Radiation Research, Lund University, Box 118, SE-221 00 Lund, Sweden. [2]NanoLund, Lund University, Box 118, SE-221 00 Lund, Sweden. [3]MAX IV Laboratory, Lund University, Box 118, SE-221 00 Lund, Sweden. [4]Physics Department, University of Trieste, via A. Valerio 2, Trieste 34127, Italy. ✉e-mail: ulrike.kust@sljus.lu.se; jan.knudsen@sljus.lu.se

pulses of $CH_4$ and $O_2$ with and without CO added were used to demonstrate that CO oxidation inhibits $CH_4$ oxidation.

Oxidation studies in methane-rich gas environments are more rare in the literature[23–25]. Here, partial methane oxidation to CO and complete oxidation to $CO_2$ are observed alongside complete decomposition of methane to carbon. The deposition of carbon has been reported to deactivate the catalyst[20,21,25], however, recent studies found that cycling between oxygen-rich and methane-rich gas compositions can prevent or significantly slow down deactivation[44] as well as enhance catalytic activity[38,45]. The dynamic evolution of carbon on the catalyst surface and in its subsurface that is expected to happen at such cycling conditions has, however, not been studied in detail. This is surprising since these dynamics likely are heavily involved in controlling whether carbon deactivates or enhances catalytic activity. Hence, understanding the delicate interplay between what triggers surface carbon formation and subsurface dissolution and how this changes catalytic activity and selectivity and how this again changes the catalyst composition is of crucial importance. To shed light on this, we modulate the Pd catalyst temperature, both to slow down catalyst deactivation that is otherwise observed in the methane-rich gas mixture and to modulate the catalytic behavior. When the temperature is high all oxygen is consumed, and we reach the O mass transfer limit (O-MTL) resulting in carbon deposition. Opposite, when the temperature decreases less oxygen is consumed and we tip out the O-MTL, moving towards complete oxidation of methane. These resulting changes in catalytic function due to a modulated temperature make it possible to not only study activity changes of the catalyst, i.e., whether more or less $CO_2$ is being produced during the methane oxidation reaction, but also to study changes in selectivity, i.e, whether CO or atomic carbon is produced instead of $CO_2$. Using a single technique, time-resolved Ambient Pressure X-ray Photoelectron Spectroscopy (tr-APXPS) coupled with mass spectrometry to detect $H_2$, we provide a complete time-resolved and quantified picture of methane conversion to $CO_2$, CO, $H_2O$, $H_2$, and surface carbon which makes it possible to directly follow catalytic activity and selectivity. Simultaneously, we follow carbon removal from the surface either via $CO_2$ formation or via dissolution in the subsurface.

In the present study we find that the changes in both activity and selectivity are related to surface and subsurface carbon. We demonstrate that the surface carbon coverage correlates with the CO production. Subsurface carbon is, however, equally important and it is by no means a passive spectator. It both controls the total methane turnover—as carbon segregation to the subsurface makes new sites available for methane activation—and changes the catalytic selectivity by indirectly controlling whether surface hydrogen produced by methane decomposition is oxidized to water or combines to $H_2$.

## Results

### Surface Spectra

To study surface deposition and bulk dissolution of carbon, we acquired spectra with two different electron kinetic energies corresponding to probing depths of 0.6 nm and 1.5 nm (3 and 8 atomic layers), respectively (c.f. Supplementary Methods for details). Figure 1a–d show image plots of the O 1s and C 1s spectra for the two probing depths measured with 8 Hz acquisition frequency. To improve the signal-to-noise ratio, we use so-called Fourier analysis[42,46] on the full datasets 1200 s long each (c.f. Supplementary Methods for details). The improved data quality (c.f. Panels e–h) allows analysis of the O 1s spectra, and thus makes it possible to track adsorbed oxygen on the surface, which was previously not possible with such a high level of confidence[23]. Examples of the curve fits of the Fourier analysis improved spectra are shown in Fig. 1i–l for two different times. Peak assignment of the surface components is based on the values reported in literature[47–50] and given in the SI. The apparent binding energies of the gas phase components, i.e., $CO_2$, $H_2O$, and $CH_4$ cannot be tabulated as their value also depends on the local surface work function. Since the latter can change during a measurement the gas phase apparent binding energies change which is observed in Fig. 1. Additional data on, for instance, the Pd 3d region, an additional probing depth of 4 nm, and the Fourier transforms is shown in Figs. S1, S2, and S3.

The curve fitting in Fig. 1k, l reveals appearing and disappearing surface (284.5 eV) and subsurface (284.9 eV) carbon components. Notice how the relative height of the subsurface carbon component - as expected - is significantly enhanced for the 1.5 nm probing depth. In contrast, only one oxide component at 529.7 eV is observed in Panels i, j in addition to the Pd $3p_{3/2}$ and the gas phase components. As the oxide component is significantly reduced when the probing depth is increased from 0.6 nm to 1.5 nm (see Panels i, j) it must be assigned to Pd oxide on the surface. A more detailed discussion of the surface spectra including time-resolved Pd $3d_{\frac{5}{2}}$ and Pd $3p_{\frac{3}{2}}$ is given in the Supplementary Discussion and Figs. S7 and S8.

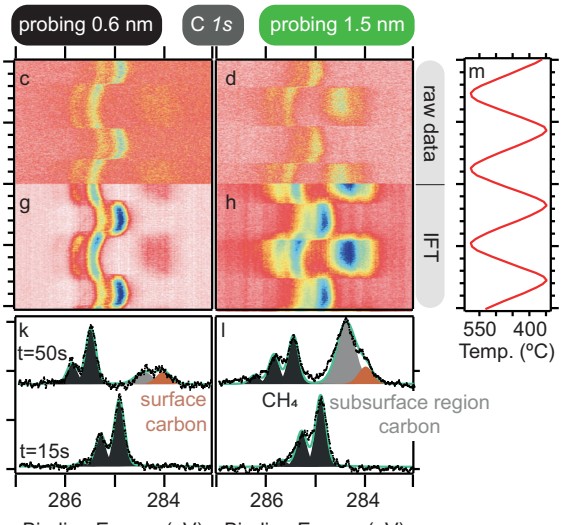

**Fig. 1 | Surface and subsurface O 1s and C 1s core level spectra acquired to follow carbon buildup and removal while modulating the temperature.** O 1s and C 1s raw data (**a–d**), de-noised spectra with Fourier analysis (**e–h**), and examples of the curve fit to the IFT at two times (**i–l**) are shown together with the temperature modulation (**m**).

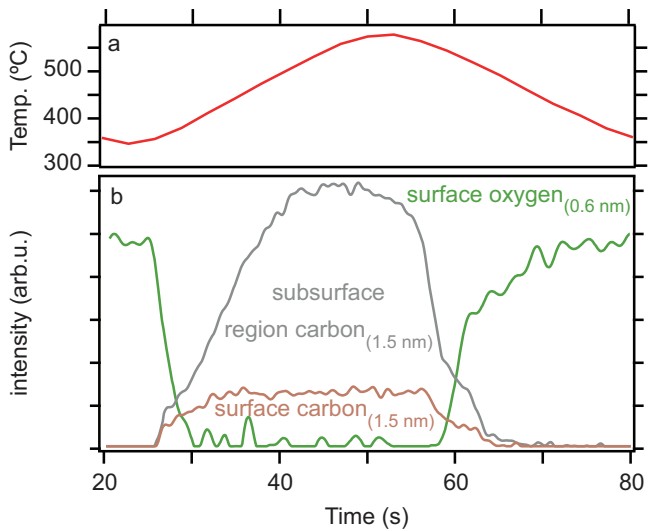

**Fig. 2 | Evolution of XPS curve-fitted surface oxygen, carbon, and subsurface carbon.** The catalyst temperature (**a**) is shown alongside the curve-fitted intensities (**b**) obtained from the IFT-improved O *1s* and C *1s* spectra shown in Fig. 1e and h, respectively.

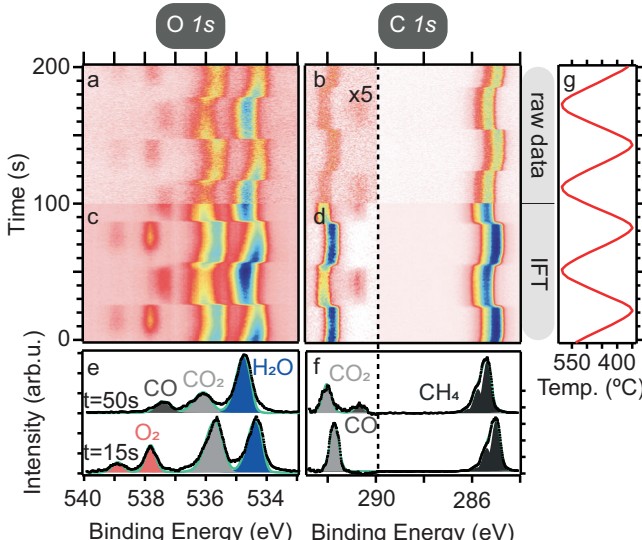

**Fig. 3 | Gas phase XP spectra following reaction products acquired while modulating the temperature.** Measured raw data for the O *1s* (**a**) and C *1s* (**b**) gas phase spectra. The respective inverse Fourier transforms (IFT) of 50 harmonics and the 0 Hz component are shown in (**c**) and (**d**). Panel (**g**) shows the temperature modulation signal applied to the catalyst. Examples of the curve fitting are shown at $t = 15$ s and $t = 50$ s in Panels (**e**) and (**f**). The high BE side of the C *1s* spectra has been magnified 5 times for better visibility.

Turning now our attention to the time evolution of the surface, Fig. 2 shows the surface and subsurface carbon (obtained from curve fitting the 1.5 nm dataset) together with the intensity of surface oxygen (obtained from the 0.6 nm dataset) as function of time alongside the catalyst temperature.

Comparing the time evolution of surface and subsurface carbon in Panel (b) it is evident that both components initially increase and finally decrease in parallel. In contrast, their time evolution between $33\,s < t < 40\,s$ is vastly different as the surface carbon here reaches a plateau while the subsurface carbon steadily increases. We explain these observations by having an equal deposition and removal rate of carbon to and from the surface. A likely scenario in this regime is that carbon is deposited once available sites for C-deposition are created by removal of carbon to the subsurface. In this scenario, the overall rate limiting step for the total carbon uptake is the dissolution of C in subsurface layers. These subsurface layers are afterwards, however, fully emptied of carbon. An example of a situation in which carbon is not fully removed from deeper catalyst layers is shown in C *1s* data measured with a slightly different temperature profile (cf. Fig. S6).

## Gas phase spectra

Gas sensitive O *1s* and C *1s* tr spectra shown in Fig. 3 are acquired and treated in the same way as and synchronized to the surface spectra discussed above. The Fourier transform data is shown in Fig. S4. Inspection of Panel (c) reveals a sudden absence of $O_2$ around $t = 28\,s$. Comparison with Fig. 1 shows that this absence of gaseous $O_2$ coincides with the absence of surface oxide. The lack of both gaseous and surface oxygen after $t = 28\,s$, signals that the reaction here enters the oxygen mass transport limit (O-MTL). This directly affects the reaction products being produced. For example, the production of CO is only observed in the O-MTL in Fig. 3 c, d.

In Fig. 4b, we plot the partial pressures calculated from the intensity evolution of the curve fits to the O *1s* and C *1s* spectra and the measured total pressure (c.f. Supplementary Methods for details) together with the temperature (a). Only the $CO_2$ and CO partial pressures calculated from the O *1s* fit are displayed as those from the C *1s* fit are similar (c.f. Fig. S5). Simultaneously, we used mass spectrometry (MS) to probe the gas composition above the catalyst probed by the nozzle of the electron analyzer (see panel c). Based on the observations of carbon deposition (Fig. 2) and CO, $CO_2$, $H_2O$, and $H_2$ production (Fig. 4b, c) we formulate a set of reaction pathways in Fig. 4e. Here,

reaction (i) is the combustion of surface carbon expected to happen in highly oxygen-rich conditions, while (II–V) describe the turnover of one methane molecule in increasingly oxygen-lean conditions. A thorough discussion of why exactly these reaction pathways are included is given in the SI.

Using these reaction pathways and the partial pressures of the respective components it is possible to determine the time-resolved selectivity (see Supplementary Discussion for a description of the exact calculation). The quantified time evolution for each reaction pathway is shown as a stacked plot in Fig. 4d. Here, the area of each color is proportional to the number of methane molecules converted via the corresponding reaction pathway.

Entering the O-MTL at $t = 28\,s$, the selectivity shifts away from complete oxidation of surface carbon and methane (reaction (II)). During the O-MTL ($28\,s < t < 60\,s$), the methane turnover changes significantly and, being limited by the supply of oxygen, the catalyst selectivity (see Fig. 4e) changes accordingly. For instance, an increasing amount of carbon is being deposited via V during the first half of the O-MTL while reaction IV is observed in the second half and after exiting the O-MTL.

Naturally, only when sufficient oxygen is available in front of the surface, carbon can be re-oxidized via reaction I. Here, carbon segregates from the bulk to the surface and is there rapidly oxidized to $CO_2$ making its residence time on the surface too short to be detected. The re-oxidation pathway is, however, a minority reaction channel as most of the deposited carbon remains buried in the bulk. Evidence for this comes from comparing the areas of the re-oxidation pathway (I) and carbon deposition (IV,V). Doing this, we find that 9% of all deposited carbon is removed via re-oxidation while the remaining 91% are dissolving in the catalyst bulk.

## Discussion

Having discussed the time-evolution of surface and subsurface carbon as well as activity and selectivity towards the different reaction channels I–V we now discuss their mutual correlations in Fig. 5.

Starting with correlations between surface composition and catalytic selectivity—which is common to search for in catalysis studies—

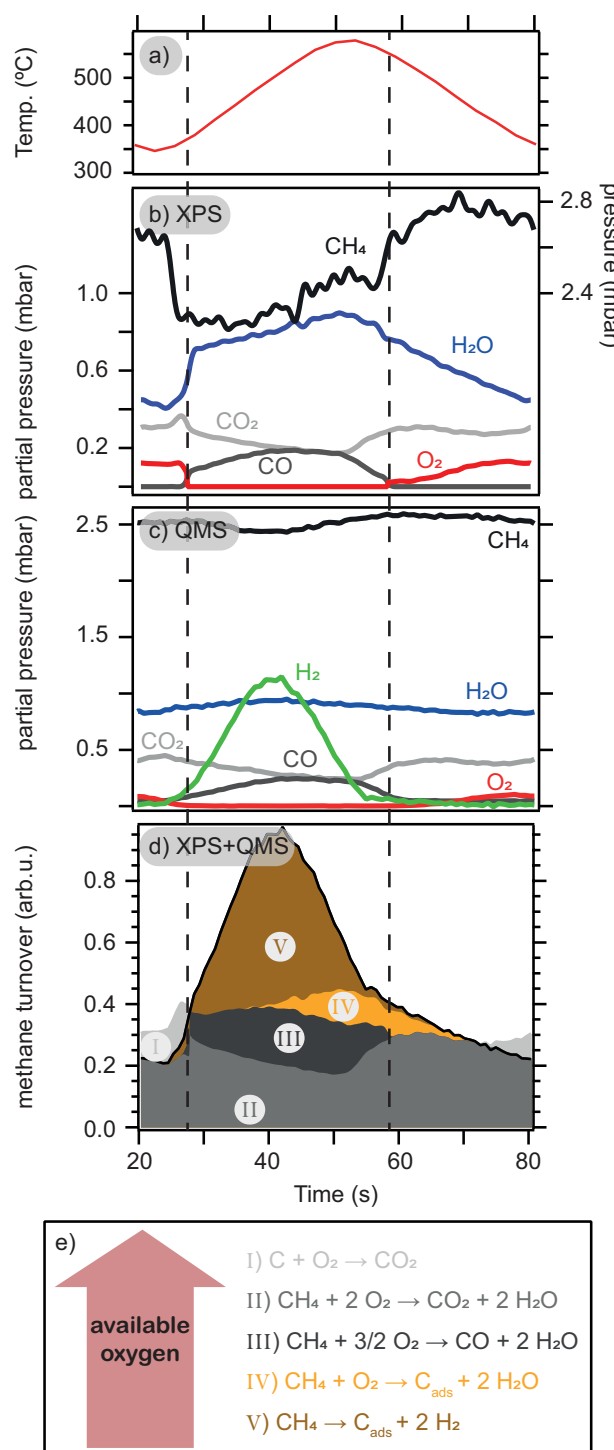

**Fig. 4 | Evolution of reactants and turnovers of different reaction pathways.** The catalyst temperature (**a**) is shown alongside the partial pressures calculated from the APXPS curve fit (**b**) together with those calculated from the MS (**c**). The time-resolved selectivity (c.f. (**e**)) within the total conversion is calculated and shown as a stacked plot (**d**) in which the colors refer to the reaction equations in (**e**). The vertical dashed lines indicate the beginning and end of the O-MTL.

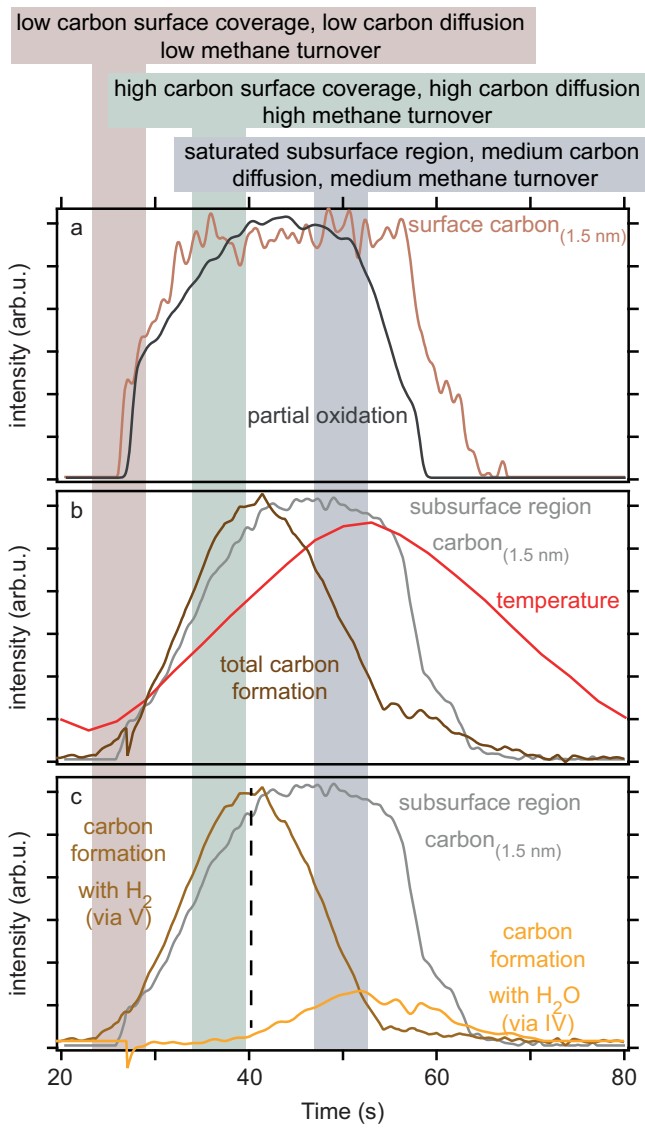

**Fig. 5 | Comparison of surface species coverages with catalytic activity and selectivity.** Comparison of surface and subsurface carbon evolutions together with reaction pathway selectivities (**a**–**c**). The evolutions are taken from Fig. 2 for the surface and subsurface C-components and from Fig. 4 for the selectivity.

Fig. 5a demonstrates that the amount of $CH_4$ converted via partial oxidation to CO (reaction (III)) correlates with the surface carbon coverage. The reason for this correlation is that a higher surface carbon coverage makes it more likely for two oxygen atoms, formed from $O_2$ dissociation, to react with one carbon atom each, forming two CO molecules, rather than reacting with the same carbon atom, forming one $CO_2$ molecule.

Going beyond the common surface structure ↔ catalytic function analysis, we can compare the subsurface carbon evolution with the catalytic activity. Doing this in Fig. 5b, we indeed observe a correlation between the growing subsurface carbon concentration and the total carbon deposition, which is given by the sum of reactions (IV) and (V) (c.f. Fig. 4), that cannot be accounted for by either the surface carbon or the temperature. This correlation suggests that increased diffusion at increased temperatures controls the amount of carbon that can be deposited. The likely reason is that diffusion channels open into the catalyst bulk, removing more carbon from the surface. This diffusion then opens up adsorption sites for methane on the surface which, in turn, results in more carbon deposition. At $t = 40$ s, however, the subsurface region is saturated with carbon limiting diffusion into the bulk similar to a traffic jam situation. That reduces the available adsorption sites and results in decreasing carbon deposition. That is, the C diffusion in the subsurface layers strongly influences the catalytic activity.

The subsurface carbon, however, not only influences catalytic activity but also selectivity. In Fig. 5c we compare the two different reaction pathways leading to carbon formation (IV,V) together with the

subsurface carbon concentration. We observe that at $t = 40\,s$, i.e., when the subsurface C concentration reaches saturation, the selected carbon formation pathway shifts away from reaction (V) toward reaction (IV). A likely reason for this is the reduced diffusion of carbon into the bulk due to the traffic jam, reducing the methane turnover and, thus, making more oxygen available. This oxygen can then react with the molecular hydrogen from reaction (V) forming water, resulting in reaction (IV). Hence, the subsurface C concentration even influences the catalyst selectivity.

Unfortunately, the effect of hydrogen diffusion on the catalytic reaction or the carbon diffusion cannot be discussed as hydrogen cannot be detected in APXPS.

To summarize, we simultaneously studied the time evolution of five different methane oxidation or decomposition pathways together with the corresponding Pd catalyst surface and subsurface region. Correlation of surface coverage ↔ subsurface concentration ↔ catalytic selectivity ↔ catalytic activity results in the finding that CO production is directly dependent on the surface carbon coverage while the overall methane turnover is controlled by the subsurface diffusion of carbon. Furthermore, we showed that the subsurface layers not only influence catalytic activity but also the selectivity of methane oxidation as the selectivity shifts from methane decomposition to oxidative decomposition once the C-diffusion limit in the subsurface layers is reached.

This study underlines the crucial role of subsurface carbon on surface reactions and the catalytic activity and selectivity. We envision that similar methodology can be used to study how the subsurface diffusion can be modified by alloying or nano-particle formation and how this, in turn, alters the catalytic activity and selectivity.

## Methods
### Experimental details
The presented data were measured at the HIPPIE beamline at the MAX IV laboratory, Sweden[51]. At the solid-gas interface branch we exposed a polycrystalline Pd catalyst to temperature pulses in a methane-rich gas environment at a total pressure of $3.9 \pm 0.1$ mbar. The gas flows were 3.5 sccm $CH_4$ and 0.5 sccm $O_2$, corresponding to 0.49 mbar oxygen and 3.41 mbar methane in the reaction mixture, while the temperature was periodically ramped from 345°C (30 s) to 580°C (30 s) with an IR Laser and a heating rate of $10°Cs^{-1}$. These conditions are known to drive the system into and out of the O-MTL triggering the onset and cease of carbon deposition[23]. With continuous acquisition of APXP spectra with a frequency of 8 Hz the catalyst response to 40 temperature pulses was recorded. More details of the experimental setup can be found in the SI.

### Measurement positions
The experiment uses surface sensitive and gas phase sensitive measurement positions. The former is defined by the overlap of the X-ray light and the analyzer focus which is fixed ≈ 600 μm away from the cone of the electron analyser. By adjusting the sample-cone distance the sample is positioned in this focus point by maximizing the surface signal. In the gas phase measurement position the sample is retracted by additonal 400 μm hereby measuring a gas phase signal with little surface background, thus increasing the relative gas phase signal and sensitivity (due to an increased beamline slit) significantly. The environmental conditions were the same for both measurement positions.

### Data analysis
All data analysis was performed in Igor Pro 8. This includes normalization to changing electron transmission through the gas phase, removal of the secondary electron background, and calibration of the binding energy (BE) axis using the Fermi edge. Subsequently, Fourier analysis[42,46] is done to improve the signal-to-noise ratio enough to enable especially curve fitting of the O 1s core level. A detailed description of this data treatment can be found in the SI.

### Depth profiling
For the most surface sensitive measurements with an approximate probing depth of 0.6 nm, we used photon energies of 450 eV (C 1s), 500 eV (Pd 3d), and 700 eV (O 1s) resulting in photoelectron kinetic energies of roughly 165 eV. For a probing depth of 1.5 nm we used 950 eV, 1000 eV, and 1200 eV, respectively, which resulted in a photoelectron kinetic energy of ~665 eV. Finally, a probing depth of ~4 nm is reached using photon energies of 1750 eV eV, 1800 eV eV, and 2000 eV eV, respectively, with an approximate photoelectron kinetic energy of 1465 eV.

## Data availability
All datasets generated and analyzed during the current study are available from the corresponding authors upon request, since private communication is necessary for the raw data analysis.

## Code availability
All codes written during the data analysis of the current study are available from the corresponding authors upon request, since private communication is necessary for the raw data analysis.

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

## Acknowledgements

We acknowledge MAX IV Laboratory for time on the beamline HIPPIE under proposal 20231733. Research conducted at MAX IV, a Swedish national user facility, is supported by the Swedish Research council under contract 2018-07152, the Swedish Governmental Agency for Innovation Systems under contract 2018-04969, and Formas under contract 2019-02496. U.K. and J.K. acknowledge financial support from Swedish Research Council (2022-04363 J.K., 2017-04840 J.K.) and the Crafoord Foundation. A.N. acknowledges financial support from PNRR DM 351 Investimento 4.1 and from Unione Europea - Next Generation EU through projects, PRIN2022 XXJNRS 2DOrNotToBe, and PRIN PNRR P2022B3WCB 2Dgo3D. We acknowledge Jason F. Weaver for fruitful discussions and suggestions for improvement when preparing this manuscript.

## Author contributions

Experiments were conducted by U.K., R.J., J.P., A.N., M. S., A.S. and J.K. U.K. analyzed the data. J. K. and A. S. supervised the project. U. K. and J. K. wrote the manuscript with input from all authors. U. K. and J. K. developed the experimental idea. All authors contributed to the discussion of the results and the manuscript.

## Funding

## Competing interests

The authors declare no competing interests.
