## [Transparent Peer Review file · Nature Communications]

Carbon Subsurface Traffic Jam as Driver for Methane Oxidation Activity and Selectivity on Palladium Surfaces

Corresponding Author: Dr Jan Knudsen

Version 0:

Reviewer comments:

Reviewer #1

(Remarks to the Author)

By applying ambient pressure XPS, the group studied the surface and subsurface of Pd under the reaction of methane oxidation in which methane is in rich. Generally, it is not clear to me the novelty of this work. If it is the depth dependent APXPS, it has been developed and used by several groups. If it is the new reaction mechanism for partial oxidation of methane, it is reported that the surface carbon will get involved. Regarding "the subsurface carbon controls the overall methane turnover", I cannot see such correlation from the data (See the following questions). In addition, there is little discussion about previous work on partial methane oxidation. Cannot tell what new thoughts this work brings to the community. Based on these, I do not recommend publishing this work at Nature Communications.

1. In Fig.1 (I-L), there is shift of peaks with the change of temperature/time even for gas phases. These shifts are real or because the peak positions are not corrected?
2. In Fig.1 (I, J), when the probing depth is 1.5 nm, the gas phases disappeared. Is that expected?
3. I don't understand this statement on Page 4: "Comparison with Fig. 1 shows that this absence of gaseous O₂ coincides with the absence of surface oxide. The lack of both gaseous and surface oxygen after $t = 28$ s, signals that the reaction here enters the oxygen mass transport limit (O-MTL). This directly affects the reaction products being produced. For example, we only observe the production of CO in the O-MTL in Fig. 3 (C,D)." When there is no O₂ and Pd oxide species, the spectra (Fig. 1I top and Fig.3E top, 3F top) show the signal of CO₂ and H₂O. Why did the author claim that the only observed product is CO?
4. In Fig. 4, based on the distribution of reactants and products at different temperatures, I agree with the authors that the reaction pathways I, II, III and V could occur. How do you know the reaction IV is involved?
5. On Page 6, I don't agree with this observation: "Even within the O-MTL ($28 \text{ s} < t < 60 \text{ s}$), no stagnation of methane turnovers are observed. Rather it continuously changes and.....". As you can see from Fig. 4b, if you think there is no stagnation of methane, then there is no stagnation of any other products.
6. On Page 7, "Also, an increasing amount of carbon is being deposited via V during the first half of the O-MTL while reaction IV is observed in the second half and after exiting the O-MTL." Could you please explain why there are two halves for carbon formation and why reaction IV is in the second half? It is not clear to me how the raw data suggests the existence of reaction IV in the second half. If it is because of the water formation, water can also be formed along with the formation of CO₂ or CO.
7. On Page 7, "only when sufficient oxygen is available in front of the surface, carbon can be re-oxidized via reaction I. This is, however, a minority reaction pathway as most of the deposited carbon remains buried in the bulk." I think this claim contradicts with the experimental observations. From XPS data, at lower T, when O₂ gas phase peak shows up, no carbon signals can be observed. Any experimental evidence showing the existence of bulk C by increasing the probing depth?
8. In Fig.5D-F, the temperature indicator does not tell me anything. Please specify the temperature range or time range for each schematic so that one can better understand each regime and associated reactions.
9. From what I see from Fig.5A is that the appearance of surface carbon comes along with the formation of CO. But in Fig. 5D, CO₂ is the only product even though there is surface carbon. For Fig. 5E, the product CO₂ is not included. But from Fig.3, you can see CO₂ is always there. For Fig.5F, how come the conversion is low at the highest temperature? As you can see from Fig.4A,B, at the highest temperature, the methane concentration is the lowest. Only Fig.5F contains water molecules which should also be there in Fig.5DE.
10. Technically, what is the purpose of performing temperature modulation? You will not be able to see CO₂, CO, methane, and carbon if staying at a certain temperature for a longer time?
11. It is known that C and H can diffuse into Pd to form hydride and carbide. Since H₂ is generated under the reaction, how does that affect C diffusion?

Reviewer #2

(Remarks to the Author)

This work combines time-resolved XPS with mass spectrometry data to detect surface carbon formation and its subsurface dissolution. A valid quantitative approach is proposed, which allows correlating carbon dissolution with the overall selectivity of methane oxidation with temperature. Results are promising and the method is interesting. However, below I list some comments that must be addressed before reconsidering the manuscript for publication.

1) Sample preparation (SI). The authors write that the sample "...was put through several oxidation and reduction cycles that roughened the surface, and,...., the surface eventually turned polycrystalline". Was the sample investigated by LEED? Why using a single crystal and not a foil?

2) Page 2, line 8. By definition, a reaction intermediate is a chemical species formed transiently during a reaction. I suggest to replace "are hydrogen, oxygen..." with "involve hydrogen...".

3) Page 3, discussion of Figure 1. The line shape of Pd 3p in Figure 1I changes a lot (FWHM and peak positionnegative binding energy shift). This probably happens in response to surface PdO formation. Can the authors expand the discussion? The same does not happen in Figure 1J. It could be due to the larger probing depth, although the signal should always contain information from the surface.

4) Page 3-4. In the methods it is reported that the reaction mixture is created dosing 3.5 sccm CH₄ and 0.5 sccm O₂. What is the partial pressures ratio between CH₄ and O₂ (based on Figure 4, it seems that measurements were performed in excess of methane)? Why has such a ratio been selected? Have experiments been performed at different CH₄/O₂ (this influences a lot chemistry at the surface)?

5) Page 4, line 10. Here it is not clear how the authors disentangled carbon dissolution from carbon segregation (for example, coke formation is well known to deactivate catalysts in methane oxidation/dry reforming). Coke contains different carbon species, which may display similar binding energies as those of surface and subsurface carbon. I suggest to show also Pd 3d, highlight changes as a function of temperature (as in Figures 1 and 2) and compare them with those of C and O 1s. I expect a different attenuation of Pd 3d in the case of dissolution/segregation, which may support the discussion. Based on Figure S2, it seems that bulk PdO is formed together with surface PdO. If so, why isn't lattice oxygen observed in the O 1s spectra? This demonstrates that data have to be thoroughly discussed.

6) Page 5, Figure 3. While in Figure 1 the signal of gas phase methane is correctly fitted with a triplet, in Figure 3 it appears (and is fitted) as a singlet. How can it be?

7) Page 6-7. Reaction I requires methane adsorption and dehydrogenation. From the literature, it is known that PdO is the active phase for low temperature methane oxidation, with methane adsorption taking place on palladium while neighbor oxygens extract hydrogen forming hydroxyls. According to this and to reaction I, hydroxyls should be detected (O 1s) at "low" temperature. I suggest to investigate (and discuss) more the O 1s - Pd 3p spectral region.

8) Page 8, line 23. Here it is not entirely clear what limits the diffusion of carbon within the metal. Is it possible to estimate the thickness of such a "diffusion layer"? Does palladium carbide form? A depth profile analysis combined with fitting using a proper attenuation function may give the thickness and carbon concentration.

9) Page 8 (GENERAL COMMENT): results discussed here are quite interesting. However, measurements are always performed in the absence of water (dry conditions). It has been demonstrated that water strongly influences the reaction (active sites poisoning). I am wondering what would be the effect of water addition to the dissolution of carbon/to the overall selectivity. Considering the effect of water would certainly increase the impact of this work.

Version 1:

Reviewer comments:

Reviewer #1

(Remarks to the Author)

This work studied a model system and a model catalyst for the purpose of taking advantage of a technique (ambient pressure XPS) and focus only on carbon species, which narrows the interest of this work. This work was performed and written as if carbon species was the decisive and only parameter controlling the selectivity and methane turnover. The dynamic change of carbon must be correlated with Pd. In addition, H can also easily diffuse into Pd, which is well known but cannot be probed by AP-XPS so it is not investigated in this work. This work can be greatly improved if acknowledge the complexity of a catalytic system and expand the investigation tools. At the current stage, my overall feeling is that this work reports a result a specific technique can provide. It can provide some insights into the carbon effects on methane oxidation reaction but more fits a specific journal.

Reviewer #2

(Remarks to the Author)

The authors did an excellent job and adequately addressed all my comments. I hope that they will keep investigating this interesting reaction making use of such advanced tools. I support the publication of this manuscript in Nature Communications

During the revision of the manuscript we changed some of the wording to improve readability. In the revised manuscript these changes have been marked.

Reviewer #1 (Remarks to the Author):

By applying ambient pressure XPS, the group studied the surface and subsurface of Pd under the reaction of methane oxidation in which methane is in rich. Generally, it is not clear to me the novelty of this work. If it is the depth dependent APXPS, it has been developed and used by several groups. If it is the new reaction mechanism for partial oxidation of methane, it is reported that the surface carbon will get involved. Regarding “the subsurface carbon controls the overall methane turnover”, I cannot see such correlation from the data (See the following questions). In addition, there is little discussion about previous work on partial methane oxidation. Cannot tell what new thoughts this work brings to the community. Based on these, I do not recommend publishing this work at Nature Communications.

We acknowledge the criticism of the reviewer and realize that we should have stated the novelty of the work more clearly in the manuscript. We have rewritten the introduction substantially, added a discussion of the previous literature both on methane oxidation (l. 30 – 58) and on the use of pulsed conditions during catalytic operation (l. 39 – 58) and point now more directly at our main findings (l. 61 – 65, l. 77 – 83).

In the paper we do not explicitly say what is “novel” as this should be avoided referring to the style and formatting guide of Nature Communication. Here we can, however, explain directly what we find novel/interesting, which reviewer 2 also acknowledges. We demonstrate the importance of subsurface carbon for CH₄ oxidation. Often, it is assumed that only the surface is responsible for the catalytic conversion which is, however, not necessarily true anymore as soon as subsurface species are involved. By following the individual methane oxidation pathways together with the evolutions of surface and subsurface carbon concentration in a time-resolved way – which has not been done previously - we demonstrate that the surface carbon coverage correlates with the CO production while subsurface carbon controls both the total methane turnover - as carbon segregation to the subsurface makes new sites available for methane activation – and the catalyst selectivity by indirectly controlling whether surface hydrogen (produced by carbon deposition) is oxidized to water or recombines to H₂. Clearly, the method used to deconvolute the effect of surface and subsurface species is not limited to carbon and it can easily be used also for example to study surface/subsurface oxides.

We hope that our changes to the manuscript makes the evidence for the correlation between the subsurface carbon concentration and methane turnover more clear.

1. In Fig.1 (I-L), there is shift of peaks with the change of temperature/time even for gas phases. These shifts are real or because the peak positions are not corrected?

The shift in the gas phase apparent BE is real and caused by a changing work function of the sample surface. This changes the potential in the near vicinity of the surface (where the x-ray beam probes the gas phase molecules) causing all gas phase peak BEs to change by the same amount as the potential in front of the sample changes. In contrast, XPS peaks of surface or subsurface elements or adsorbates are unaffected by work function changes as the sample is grounded. These atoms are therefore positioned in a ground potential.

The change of gas phase peaks due to work function changes of the sample surface has been observed before, see for example in Knudsen et al. *ACS Catalysis*, 15 (3), 1655-1662 (2025), Teschner et al. *The Journal of Physical Chemistry C*, 128 (17), 7096-7105 (2024), or Küst et al. *ACS Catalysis*, 14, 5978–5986 (2024) and can be used to differentiate surface from gas phase species as well as to indirectly observe small changes at the surface.

We realize, however, that we should have explained this more carefully in our article for the general readership of Nature Communication and added a more detailed explanation in the revised manuscript (l. 96 – 99).

2. In Fig.1(I, J), when the probing depth is 1.5 nm, the gas phases disappeared. Is that expected?

The reason for the higher probing depths is that we used higher photon energies. This leads to a higher electron kinetic energy which leads to a higher electron mean free path. Thus, the detected electrons can originate from deeper in the bulk. At the same time, the amount of gas and surface components probed remains the same. Thus, surface and gas component intensities are reduced in comparison. However, they can still be observed for example in panel J at high binding energies.

The reason why the CH₄ signal is so much stronger as compared to the bulk carbon signal in Panel L is the methane concentration in front of the surface is rather high while the concentration of bulk carbon is not that high. In contrast, the oxygen concentration in front of the surface is rather low while there are many Pd bulk atoms being probed.

In the revised manuscript, we explain that gas phase components change in response to work function changes (l. 96 – 99) and we explain why the gas phase peaks decrease in intensity at higher probing depths in the SI in the extended discussion of the surface spectra (l. 133 – 140).

3. I don't understand this statement on Page 4: "Comparison with Fig. 1 shows that this

absence of gaseous O₂ coincides with the absence of surface oxide. The lack of both gaseous and surface oxygen after t = 28 s, signals that the reaction here enters the oxygen mass transport limit (O-MTL). This directly affects the reaction products being produced. For example, we only observe the production of CO in the O-MTL in Fig. 3 (C,D).”

When there is no O₂ and Pd oxide species, the spectra (Fig. 1I top and Fig. 3E top, 3F top) show the signal of CO₂ and H₂O. Why did the author claim that the only observed product is CO?

We realize that we have phrased the observation rather poorly. We did not mean that CO is the only reaction product. Instead, we wanted to describe that the specific reaction product CO is only observed when the reaction is mass transfer limited by oxygen. In contrast, CO₂ and water are being produced both in- and outside the O-MTL. In the revised manuscript we reformulated the sentence (l. 126 – 127) from “For example, we only observe the production of CO in the O-MTL in Fig. 3 (C,D).” to “For example, the production of CO is only observed in the O-MTL in Fig. 3 (C,D)”.

4. In Fig. 4, based on the distribution of reactants and products at different temperatures, I agree with the authors that the reaction pathways I, II, III and V could occur. How do you know the reaction IV is involved?

Inspection of fig. 4E shows that each methane turnover generates two H₂ or two H₂O molecules. As we measure in a flow reactor and continuously remove formed products, the total methane turnovers (black solid convoluting curve in fig. 4D) are proportional to the sum of $2 \times (p(\text{H}_2) + p(\text{H}_2\text{O}))$. Methane turnovers to CO (channel III in fig. 4E) are proportional to $p(\text{CO})$ and methane decomposition turnovers are proportional to $p(\text{H}_2)/2$ (reaction V) again referring to panel E. The remaining H₂O not produced via channel III can be calculated as $(p(\text{H}_2\text{O}) - 2p(\text{CO}))$. This remaining H₂O comes from either reaction II in oxygen-rich conditions or a combination of reaction II and reaction IV in methane-rich conditions. Simultaneously, the CO₂ production can be balanced either by pathway I and II for O₂-rich conditions or solely by reaction pathway II for CH₄-rich conditions. The balancing factors for each reaction pathway that are needed to make everything match up directly give the turnovers for channel I, II, and IV.

In the revised version of the SI we included both the above discussion which we believe is easier to understand for non-experts as well as a more mathematically accurate formulation of this reasoning (l. 83 – 97). After careful consideration we decided not to include this in the manuscript itself, partly because it is very technical, and partly because this kind of analysis is not new and has been discussed before in Küst et al. *ACS Catalysis*, 14, 5978–5986 (2024).

5. On Page 6, I don't agree with this observation: "Even within the O-MTL ($28 \text{ s} < t < 60 \text{ s}$), no stagnation of methane turnovers are observed. Rather it continuously changes and.....". As you can see from Fig. 4b, if you think there is no stagnation of methane, then there is no stagnation of any other products.

We think that it came to a misunderstanding here. In CO oxidation for example, if the supply of one reactant is limited by mass transfer, the formation of CO₂ stagnates and even upon increase in temperature, not more of the other reactant can be combusted. In methane oxidation, however, even if oxygen is in limited supply, more methane can be combusted by simply choosing methane oxidation pathways that require less oxygen.

We see, however, how it could come to the misunderstanding and reformulated in the revised manuscript (l. 144 – 149) changing the sentence "Even within the O-MTL ($28 \text{ s} < t < 60 \text{ s}$), no stagnation of methane turnovers are observed. Rather it continuously changes and, being limited by the supply of oxygen, the catalyst selectivity (c.f. Fig. 4 (E)) changes accordingly." into "During the O-MTL ($28 \text{ s} < t < 60 \text{ s}$), the methane turnover changes significantly and, being limited by the supply of oxygen, the catalyst selectivity (c.f. Fig. 4 (E)) changes accordingly."

6. On Page 7, "Also, an increasing amount of carbon is being deposited via V during the first half of the O-MTL while reaction IV is observed in the second half and after exiting the O-MTL." Could you please explain why there are two halves for carbon formation and why reaction IV is in the second half? It is not clear to me how the raw data suggests the existence of reaction IV in the second half. If it is because of the water formation, water can also be formed along with the formation of CO₂ or CO.

As discussed for comment 4, if more water is produced than would be expected by the amount of CO and CO₂ that is detected, reaction IV must be present. This is the case in the second half of the O-MTL.

Now, the reason why this happens in the second half of the O-MTL is also not entirely clear to us. What we can see is that the methane turnover starts to decrease approximately in the middle of the O-MTL, making more oxygen available per 'oxidized' methane. Thus, instead of forming hydrogen and carbon (reaction V) the hydrogen can now react with some of the available oxygen forming water (reaction IV).

While working on this comment we realized that renaming the y-axis of Fig. 4 (D) from partial pressures to methane turnover increases understandability.

7. On Page 7, "only when sufficient oxygen is available in front of the surface, carbon can be re-oxidized via reaction I. This is, however, a minority reaction pathway as most of the deposited carbon remains buried in the bulk."

I think this claim contradicts with the experimental observations. From XPS data, at lower T, when O₂ gas phase peak shows up, no carbon signals can be observed. Any experimental evidence showing the existence of bulk C by increasing the probing depth?

We don't see the contradiction here. Just because no surface carbon peak is measured does not mean that no surface carbon species exists. The species could either have a very low concentration or a very short residence time, both of which makes it undetectable in XPS. An example of this is the formation of gaseous CO during the O-MTL. The presence of CO in the gas phase requires the presence of CO on the surface. This is not detected in the experiment, however. We added a sentence to the manuscript (l. 151– 152) to avoid future misunderstandings changing “[...] carbon can be re-oxidized via reaction I. This is, [...]” to “[...] carbon can be re-oxidized via reaction I. Here, carbon segregates from the bulk to the surface and is there rapidly oxidized to CO₂ making its residence time on the surface too short to be detected. The re-oxidation pathway is, [...]”.

In the C 1s surface data measured at 4 nm probing depth (see SI Fig. S3 (F,L)), some carbon can be observed even at low temperatures. If much deeper catalyst layers should be probed, increasing the photon energy would be necessary. In this experiment, the maximum photon energy (2keV) was too low to probe much deeper bulk layers. However, further evidence for the existence of bulk C comes instead from the fact that gradual catalyst deactivation was observed. The initial activity could only be restored by heating the catalyst for an hour at 500 °C in an oxygen atmosphere. Even though the surface remained metallic, significant CO₂ production was observed during the first 30-40 min. This can only be explained by carbon being stored in the catalyst bulk.

8. In Fig.5D-F, the temperature indicator does not tell me anything. Please specify the temperature range or time range for each schematic so that one can better understand each regime and associated reactions.

We understand the criticism of the reviewer both for comments 8 and 9. The sketches were really just meant to schematically point out the most important differences between different time/conversion regimes. Since they caused so much misunderstandings we decided to remove them in the revised manuscript and add comments above the figure for the different regimes instead (see Fig. 5).

9. From what I see from Fig.5A is that the appearance of surface carbon comes along with the formation of CO. But in Fig. 5D, CO₂ is the only product even though there is surface carbon.

As written for comment 8, we adjusted the figure to avoid the misinterpretations.

In Panels D and E of the old figures we wanted to highlight that, for a low surface carbon coverage, CO₂ is the main reaction product while, for a high surface carbon coverage, more CO is being produced. The sketches were never meant to be quantitative but rather to qualitatively highlight the most important differences.

For Fig. 5E, the product CO₂ is not included. But from Fig.3, you can see CO₂ is always there.

We are grateful the reviewer made us aware that our old figures 5E-F could be read in different ways making it hard to follow our argumentation. As discussed above the intention with the old figure was to highlight reaction product changes rather than showing all reaction products, which was not clear. Figure 5E is removed in the revised version to avoid misunderstandings.

For Fig.5F, how come the conversion is low at the highest temperature? As you can see from Fig.4A,B, at the highest temperature, the methane concentration is the lowest.

Due to a comment of reviewer 2 who asked us to fit the methane peak with all its vibrational levels, just as done for the surface spectra, we redid the C 1s gas phase curve fit. Here, we realized that there was a mistake in the previous CH₄ time evolution that mistakenly had been overwritten by another time-evolution during the data treatment while making the final figures.

In the revised manuscript, we present the correct CH₄ time evolution which supports our conclusions as presented in the manuscript. The lowest methane partial pressure is reached at $t = 38$ s, i.e., before the temperature reaches the maximum value and coinciding with the maximum hydrogen partial pressure. We also checked all other time-evolution curves.

We do not use the methane signal for statements about the methane conversion as we do not know how much methane we would have in front of the surface with zero conversion (since we operate with an active catalyst at 350 °C – 580 °C all the time). Thus, the methane conversion proportional to the difference between the methane pressure at zero conversion minus the current methane pressure cannot be calculated. Instead, we use the water and hydrogen partial pressures in front of the surface to calculate the methane conversion. As discussed above, for the reaction pathways II-V the conversion of each methane molecule leads to the formation of either two water molecules or two hydrogen molecules. Thus, half the sum of the water and the hydrogen partial pressures is equal to the total methane turnover.

Now, this sum peaks before reaching the highest temperature. This is direct evidence for an increasing temperature when the methane conversion starts to drop. The reason for this is – as we show in the manuscript - the filling of the subsurface with carbon.

Only Fig.5F contains water molecules which should also be there in Fig.5DE.

As discussed above the intention with the old figure was to highlight reaction product changes rather than showing all reaction products, which was not clear. Figure 5F is removed in the revised version to avoid misunderstandings.

10. Technically, what is the purpose of performing temperature modulation? You will not be able to see CO₂, CO, methane, and carbon if staying at a certain temperature for a longer time?

Indeed, we will also see these products in steady state experiments. However, we use the temperature modulation also “to slow down catalyst deactivation that is otherwise observed in the methane-rich gas mixture and to modulate the catalytic behavior. When the temperature is high all oxygen is consumed, and we reach the O mass transfer limit (O-MTL) resulting in carbon deposition. Opposite, when the temperature decreases less oxygen is consumed and we tip out the O-MTL, moving towards complete oxidation of methane. These resulting changes in catalytic function due to a modulated temperature make it possible to not only study activity changes of the catalyst ...” (l. 62 – 68). Measuring time-resolved spectra makes it possible to study the transitions between an oxidized surface, a carbidized surface and the evolution of the carbon diffusion. We modulate periodically to be able to employ Fourier analysis to improve the signal-to-noise ratio of the (especially O 1s) data and make them analyzable. We appreciate, however, that this might not have been as clearly described as we had hoped and reformulate in the revised manuscript as described in the citation above. Overall, modulating environmental parameters is a much employed method called ‘Modulation Excitation Spectroscopy’ which is often used to study catalysts at off-equilibrium conditions, a fact that we added to the general discussion of the field in the introduction (l. 43 – 47) (see Müller *et al.* *Applications of Modulation Excitation Spectroscopy*, 56 (5), 1123-1136 (2017), Roger *et al.* *Chemical Science*, **14**, 7482-7491 (2023), or Urakawa *et al.* *Springer International Publishing* (2023) p. 967-977).

11. It is known that C and H can diffuse into Pd to form hydride and carbide. Since H₂ is generated under the reaction, how does that affect C diffusion?

This is a very valid comment and one of the shortcomings of the presented method. Since APXPS cannot detect gas phase hydrogen (the hydrogen electron is de-localized which makes core level spectroscopy impossible) we use mass spectrometry to provide the information about the time evolution of the hydrogen concentration. Unfortunately, we do not have the same possibility when it comes to measuring hydrogen in the crystal. Hence, we are blind to the effects of hydrogen on C diffusion or the reaction itself. We add a comment about this in the revised manuscript (l. 191 – 192) since it is important to also know about the limitations of the method.

Reviewer #2 (Remarks to the Author):

This work combines time-resolved XPS with mass spectrometry data to detect surface carbon formation and its subsurface dissolution. A valid quantitative approach is proposed, which allows correlating carbon dissolution with the overall selectivity of methane oxidation with temperature. Results are promising and the method is interesting.

However, below I list some comments that must be addressed before reconsidering the manuscript for publication.

We are happy that the reviewer finds our approach valid, our results promising and the method interesting. We thank the reviewer for carefully reading our paper and for very good suggestions to improve the paper. All comments – except one - have been addressed in the revised manuscript. We did not adhere to the last suggestion about additional wet experiments. New experiments in $\text{H}_2\text{O}:\text{O}_2:\text{CH}_4$ flows are a completely new story while the current paper already describes a self-contained study conducted in constant $\text{O}_2:\text{CH}_4$ flows while solely using temperature oscillations to modulate the reaction. Our story is focused on the changes in catalyst activity and selectivity when oxygen starts to limit the reaction. Also, the water poisoning effect reported in the literature (based on water dissociation and blocking of CH_4 adsorption sites on the oxide) is not immediately relevant for our study as we mainly study the reduced surface.

Additionally, if wet experiments and their discussion were to be added to this paper we are afraid that it will increase the complexity of the already complicated story which, we think, will not be appreciated by the general readers of Nat. Communication. Finally, additional beamtimes on wet experiments and their analysis would delay manuscript submission by at least 18 months which we would like to avoid as we strongly believe in rapid communication of scientific findings.

1) Sample preparation (SI). The authors write that the sample "...was put through several oxidation and reduction cycles that roughened the surface, and,...., the surface eventually turned polycrystalline". Was the sample investigated by LEED? Why using a single crystal and not a foil?

After accidentally destroying the single crystal by operating it in oxygen-lean conditions for too long, the polished surface was transformed into a roughened surface. The sample was investigated with LEED in which no diffraction pattern could be observed, a fact that we now mention in the revised manuscript of the SI (l. 27 – 28). Investigation with Grazing Incidence X-ray Diffraction that probes a few nm deep into the sample shows powder rings as well as Pd(100) Bragg spots suggesting that the near-surface

region is fully polycrystalline while the bulk still contains a single crystalline phase (l. 29 – 30).

The reason for using this crystal is to be consistent with our previous measurements on this sample (e.g. Küst et al. *ACS Catalysis*, 14, 5978–5986 (2024)). However, we see no reason why the experiment could not be reproduced on a foil and we added this to the discussion of sample preparation in the SI in the revised version (l. 33 – 35).

2) Page 2, line 8. By definition, a reaction intermediate is a chemical species formed transiently during a reaction. I suggest to replace "are hydrogen, oxygen..." with "involve hydrogen...".

That is a very good comment and we adapted the manuscript accordingly (l. 23 –24).

3) Page 3, discussion of Figure 1. The line shape of Pd 3p in Figure 1I changes a lot (FWHM and peak positionnegative binding energy shift). This probably happens in response to surface PdO formation. Can the authors expand the discussion? The same does not happen in Figure 1J. It could be due to the larger probing depth, although the signal should always contain information from the surface.

That is true. The peak position shift and change in width is due to the formation of oxide and carbide phases. Since the influences of the chemistry on the surface are much better visible in the less crowded and stronger Pd 3d peaks, we add a discussion of spectra at a few selected times and their curve fits to the SI (l. 148 – 160, Fig. S8 and its caption). However, similar peak shape developments as in the Pd 3d 5/2 peak can be observed in the Pd 3p 3/2 which is the one picked up in the O 1s region (see revised SI, Fig. S7).

Overall, since Pd bulk, Pd surface, PdO bulk and surface, as well as PdC bulk and surface peaks are expected, all within a range of 2 eV, the curve fit is rather arbitrary and should not be taken as sole ground for conclusions about the dynamics of the system. It can, however, support the conclusions that we could already make from the O 1s and C 1s surface spectra (see figure S8 and discussion in revised SI (l. 148 – 160)).

Even though the peak position shift in the spectra measured with larger probing depth is much smaller it can still be observed (see left-bend banana shape in Fig. S1 E). The reason why it is so much smaller is that most of the now probed Pd atoms are not exposed to the chemistry happening at the surface and, thus, are not subject to binding energy shifts.

4) Page 3-4. In the methods it is reported that the reaction mixture is created dosing 3.5 sccm CH₄ and 0.5 sccm O₂. What is the partial pressures ratio between CH₄ and O₂ (based on Figure 4, it seems that measurements were performed in excess of methane)? Why has such a ratio been selected? Have experiments been performed at different CH₄/O₂ (this influences a lot chemistry at the surface)?

As the flow controls are calibrated for the gases used, the partial pressure of methane in front of the sample is seven times higher than that of oxygen (in the reaction mixture). Since we, in our measurements, only observe the active catalyst, this ratio is never seen since some of the reactants have already been converted into products. If we were to observe the reaction mixture we would observe something close to 0.5 mbar oxygen and 3.4 mbar methane, however.

This ratio has been selected since we from our previous work know that oxygen lean mixtures of approximately this reactant ratio are needed to drive the system into the O-MTL at these temperatures and trigger carbon formation which is what we wanted to study in this work.

In the revised manuscript we added this discussion to the experimental details (l. 212 – 213).

Changing the gas composition slightly would significantly change the observed catalytic behaviour. Examples for this are all the studies previous published and performed in more oxygen rich gas environments during which only surface oxides and sometimes metallic surfaces are observed, never a carbide phase, though. We also did experiments with different temperatures or gas compositions yielding (very) different results which shows how sensitive the system is to changes in temperature/pressure/gas composition. To include additional datasets and their corresponding analysis in a communication we find difficult and fear that it will dilute the message.

5) Page 4, line 10. Here it is not clear how the authors disentangled carbon dissolution from carbon segregation (for example, coke formation is well known to deactivate catalysts in methane oxidation/dry reforming). Coke contains different carbon species, which may display similar binding energies as those of surface and subsurface carbon. I suggest to show also Pd 3d, highlight changes as a function of temperature (as in Figures 1 and 2) and compare them with those of C and O 1s. I expect a different attenuation of Pd 3d in the case of dissolution/segregation, which may support the discussion.

Our naming of the carbon-containing surface and subsurface components is not only based on binding energy shifts but also on the depth profile arguments. Additionally, to

rule out the formation of a thick coke layer, we note that we, at all times measure quite a strong Pd signal which means that Pd atoms must be present in all of the probed layers.

As mentioned above, the curve fit of the Pd 3d spectra is rather arbitrary in this case as many different peaks with only slightly different binding energies are expected. We added five examples of Pd 3d spectra and their possible curve fit in a new Fig. S8 to the revised SI. Fit parameters are based on Teschner et al. *Journal of Catalysis*, 230 (1), 186-194 (2005), Toyoshima et al. *Journal of Physical Chemistry C*, 116 (35), 18691-18697 (2012), Yue et al. *Nature Communications*, 15, 4678 (2024), Teschner et al. *Journal of Catalysis*, 242 (1), 26-37 (2006), and Teschner et al. *Journal of Physical Chemistry C*, 114 (5), 2293-2299 (2010).

The curve fit can support the conclusions made based on the O 1s and C 1s surface spectra but we do not feel comfortable to base strong conclusions solely on the curve fit of the Pd 3d data such as thickness of the PdC layer for instance. Since the Pd 3d discussion, thus, does not add much to the scientific story but rather serves as additional support for our discussion, we added the curve fits (see Fig. S8) and their discussion (l. 148 – 160) to the SI in the revised version of the manuscript.

Based on Figure S2, it seems that bulk PdO is formed together with surface PdO. If so, why isn't lattice oxygen observed in the O 1s spectra? This demonstrates that data have to be thoroughly discussed.

We are not sure what experimental observation in fig. S2 that the reviewer links to formation of bulk PdO. As mentioned earlier, due to the high number of peaks close to each other in the Pd 3d region we want to refrain from making conclusions especially if they are not supported by the O 1s or C 1s region, respectively. In the case of the PdO phase, we see clearly from the O 1s spectra that no PdO bulk phase exists.

If the reviewer correlated the shift of the Pd 3d peak to higher binding energies at high temperatures to bulk PdO, we disagree about this assignment. Instead, we assign it to the formation of a PdC phase (which can also be seen in the C 1s spectra at the same time). Unfortunately, PdC and PdO have very similar binding energies in the Pd 3d region (335.3 eV and 335.5 eV) and are, thus, hard to tell apart. We tried to illustrate this at the example of a few selected spectra together with their curve fit which we show in the SI (see Fig. S8).

6) Page 5, Figure 3. While in Figure 1 the signal of gas phase methane is correctly fitted with a triplet, in Figure 3 it appears (and is fitted) as a singlet. How can it be?

For the gas phase experiments we increased the beamline slit to 50 μm while we for surface sensitive experiments used a value of 30 μm . Additionally, the pass energy for

the electron analyzer was chosen to be 200 eV for the gas phase experiments and 100 eV for the surface experiments. Thus, the energy resolution is much better for the surface measurements than the gas measurements since both the analyzer and the beamline energy resolution is better. For this reason, the vibrational splitting of methane is much better visible in the surface than in the gas phase measurements.

Therefore, the CH₄ components in Fig. 3 are much broader and for simplicity we just fitted them with one asymmetric component including all vibrational components in the old version of our article.

We agree however that it is confusing to use vibrational components in Fig. 1 and not in Fig. 3. In the revised manuscript we included also vibrational components in the fit of Fig. 3 D. We updated also Fig. 4. Unfortunately, there was a mistake in the previous CH₄ curve fitted time evolution (we accidentally overwrote it with the water time evolution when making the final figures) which is why the result now looks different. Since we do not use the methane time evolution for any of the conclusions in the paper, however, this corrected time evolution of CH₄ has no influence on the discussions in the paper.

7) Page 6-7. Reaction I requires methane adsorption and dehydrogenation. From the literature, it is known that PdO is the active phase for low temperature methane oxidation, with methane adsorption taking place on palladium while neighbor oxygens extract hydrogen forming hydroxyls. According to this and to reaction I, hydroxyls should be detected (O 1s) at "low" temperature. I suggest to investigate (and discuss) more the O 1s - Pd 3p spectral region.

We can indeed see a small shoulder at the high binding energy side of the oxide peak in the O 1s region which might be due to hydroxyls. Unfortunately, due to the high time resolution that is essential for this study, the signal to noise ratio is rather low which is why we want to refrain from drawing too many conclusions from the spectra. We appreciate the comment, however, and added a discussion about the possibility of the existence of hydroxyl species to the SI (l. 141 – 147).:

“Hydroxyls on palladium surfaces during methane oxidation are observed at temperatures similar to those in this experiment (e.g. at 450 °C in Li *et al.*, *ACS Catalysis*, 10, 5783-5792 (2020)) and are expected to appear in the O 1s spectral region at around 531.6 eV or just between the oxide and the Pd 3p_{3/2} peak. Indeed, one can notice a small shoulder of the PdO peak in Fig. S1 (J) at higher binding energies. Unfortunately, due to the high time resolution and the resulting low signal-to-noise ratio we want to avoid drawing a conclusion from such a small difference.”

8) Page 8, line 23. Here it is not entirely clear what limits the diffusion of carbon within the metal. Is it possible to estimate the thickness of such a "diffusion layer"? Does

palladium carbide form? A depth profile analysis combined with fitting using a proper attenuation function may give the thickness and carbon concentration.

As mentioned earlier, we refrain from doing quantitative calculations as we do not think that the Pd 3d curve fit is a reliable way to follow the thickness of the diffusion layer. Instead, we pay attention to the observation that the carbon bulk signal in Fig. S3 F never fully reaches the background noise level (see figure below).

This is, however, the case for a probing depth of 1.5 nm. We conclude, therefore, that the topmost ~3 nm of the crystal are periodically filled with and emptied from carbon but that, at deeper layers, a constant non-zero carbon concentration can be observed. Since this is a very interesting point of discussion, we add it to the SI (l. 127 – 132).

“To estimate the thickness of the carbon diffusion layer, we note that the probed catalyst volume in Fig. S3 (E,K) is fully emptied of carbon at lower temperatures. When increasing the probing depth to about 4 nm, this is no longer the case (see Fig.S3 (F,L)). A constant non-zero carbon signal (at 283.9 eV) is measured indicating that, while the topmost (3 nm) catalyst surface layers are periodically filled with and emptied from carbon, the deeper catalyst layers (deeper than 4 nm) never reach zero carbon concentration.”

9) Page 8 (GENERAL COMMENT): results discussed here are quite interesting. However, measurements are always performed in the absence of water (dry conditions). It has been demonstrated that water strongly influences the reaction (active sites poisoning). I am wondering what would be the effect of water addition to the dissolution of carbon/to the overall selectivity. Considering the effect of water would certainly increase the impact of this work.

Even though the reactant mixture is dry, water is present in front of the catalyst surface due to the reaction so our experiments are in a sense not fully dry.

To add extra water is certainly an interesting idea for a future beamtime. Unfortunately, applying for beamtime in the next round (in September) would have the beamtime scheduled in spring 2026 and with the subsequent data analysis would delay manuscript submission by several months extra. This would result in a dramatic delay in

communication with the scientific community which we would like to avoid as we strongly believe in the fast communication of scientific findings. Additionally, we are afraid that the additional discussion of another experiment within the space limitations will make the already complex story even more difficult to understand for the general audience of Nat. Communications which is why we, unfortunately, cannot follow the reviewer's suggestion to include wet experiments and their discussion in this paper.